# The Association between Zinc and Copper Circulating Levels and Cardiometabolic Risk Factors in Adults: A Study of Qatar Biobank Data

**DOI:** 10.3390/nu13082729

**Published:** 2021-08-09

**Authors:** Abdelhamid Kerkadi, Dana Samir Alkudsi, Sara Hamad, Hanan Mohamed Alkeldi, Reem Salih, Abdelali Agouni

**Affiliations:** 1Human Nutrition Department, College of Health Sciences, QU Health, Qatar University, Doha P.O. Box 2713, Qatar; da1701633@qu.edu.qa (D.S.A.); sh1604803@qu.edu.qa (S.H.); ha1608292@qu.edu.qa (H.M.A.); rsalih@qu.edu.qa (R.S.); 2Pharmaceutical Sciences Department, College of Pharmacy, QU Health, Qatar University, Doha P.O. Box 2713, Qatar; aagouni@qu.edu.qa; 3Biomedical and Pharmaceutical Research Unit (BPRU), QU Health, Qatar University, Doha P.O. Box 2713, Qatar

**Keywords:** Qatar Biobank, zinc, copper, Zn/Cu ratio, cardiometabolic risk, metabolic syndrome, adults

## Abstract

Cardiometabolic risk (CMR) factors increase the likelihood of developing cardiovascular diseases (CVD). In Qatar, 24% of the total deaths are attributed to CVDs. Several nutritional disturbances have been linked to high risk of CVD. Many studies have discussed the effects of zinc (Zn) and copper (Cu) on CMR factors; however, evidence has been controversial. This investigated the association between CMR factors and the status of Zn and Cu, in addition to Zn/Cu ratio. A total of 575 Qatari men and women aged 18 years and older were obtained from Qatar Biobank. Plasma levels of Zn and Cu were determined using inductively coupled plasma mass spectrometry (ICP-MS). Anthropometric data and CMR factors were determined using standard methods. Adjusted associations between trace minerals and CMR were estimated by logistic regression. Partial correlation was performed to test the strength of the associations. Zn was not strongly correlated (*p*-value ˃ 0.01) or significantly associated with CMR factors and metabolic syndrome (MetS). Cu levels correlated positively with body mass index (BMI) (0.23; *p* ˂ 0.001), pulse rate (PR) (0.18; *p* ˂ 0.001), total cholesterol (0.13; *p* = 0.01), and high-density lipoproteins (HDL) (0.27; *p* ˂ 0.001); and negatively with diastolic blood pressure (DBP) (−0.13; *p* = 0.01). High plasma Cu significantly decreased the risk of metabolic syndrome (MetS) (0.121; *p* ˂ 0.001). Furthermore, Zn/Cu ratio positively correlated with waist circumference (0.13; *p* = 0.01), systolic blood pressure (0.13; *p* ˂ 0.01), and DBP (0.14; *p* ˂ 0.01); and negatively with BMI (−0.19; *p* ˂ 0.001), PR (−0.17; *p* ˂ 0.001), and HDL (−0.27; *p* ˂ 0.001). High Zn/Cu ratio increased the prevalence of low HDL (4.508; *p* ˂ 0.001) and MetS (5.570; *p* ˂ 0.01). These findings suggest that high plasma Cu levels are associated with a protective effect on DBP, HDL and MetS and that high plasma Zn/Cu ratio is associated with the risk of having low HDL and MetS.

## 1. Introduction

Cardiometabolic risk (CMR) factors increase the likelihood of developing cardiovascular events. The most notable of these factors include, dyslipidemia [1], hypertension (HTN) [2], central obesity [3], diabetes [4], and inflammation [2]. Cardiovascular diseases (CVD) are the number one cause of deaths with 17.9 million (31%) deaths in 2016 worldwide [5]. CVD are the cause for 34% of the MiddLe East population’s deaths. Moreover, CVD prevalence was found to be 13.7% in the region [6]. For instance, in 2016, 24% of the total deaths of all ages was attributed to CVD in Qatar [7].

Zinc (Zn) and copper (Cu) are important trace minerals in the body. Zn is a heavy metal known to suppress inflammation by regulating cytokine expression [8,9], reducing oxidative stress by activating antioxidant enzymes, and catalyze lipid metabolic enzymes [10,11,12,13]. It is also used by muscle and fat cells to metabolize glucose where it acts as a cofactor for many intracellular enzymes [14] and regulates insulin receptors’ synthesis and signal transduction mechanism [15]. Zn mediates its metabolic actions via the action of several Zn transporters that regulate Zn homeostasis and control its cellular compartmentalization [16]. For instance, hyperglycemia-mediated overexpression of Zrt/Irt-like protein-7 (ZIP7), a Zn influx transporter, in cardiomyocytes drives the activation of the inflammatory pathway of endoplasmic reticulum (ER) stress [17]. ER stress has been closely associated with the development of insulin resistance and diabetes by impeding insulin receptor signaling and dysregulating pancreatic beta cell survival that are critical factors in the pathogenesis of diabetes [18,19,20]. Furthermore, ZIP7 was shown to contribute to glucose homeostasis in skeletal muscle [21]. Skeletal muscle is an important storage site for Zn with around 60% of total body Zn content [22]. Deletion of ZIP7 transporter in a mouse myoblast cell line disturbed the expression of various genes involved in glucose metabolism and insulin receptor signaling pathway [23]. Additionally, ZIP6 and ZIP7 transporters play a key role in regulating the secretion of insulin through the changes in the sub-cellular localization of Zn pools [24]. Zn is critical for the insulin storage in granules inside beta cells by forming inactive hexamers with insulin [25]. When the Zn/insulin hexamers reach the systemic circulation, a pH-dependent dissociation of this complex enables the liberation of active monomeric insulin [26]. The Zn transporter ZnT8 is responsible for translocating Zn to pancreatic insulin granules. Genomic studies have identified that single nucleotide polymorphisms (SNP) in ZnT8 gene produce a variant protein that is associated with higher risk of type-2 diabetes (T2DM) [27]. Moreover, Zn plays a role in controlling blood pressure and vascular tone modulation through the inhibition of the transactivation ability of nuclear factor κ-light-chain-enhancer of the activated B cells (NF-κB) resulting in the regulation of the inducible nitric oxide synthase (iNOS) activity and expression [28].

On the other hand, Cu is an essential cofactor for numerous antioxidant [29] and oxidoreductive enzymes [30] which reduce atherosclerosis and inflammation, and improve cardiovascular function and fat metabolism [31,32,33,34]. Cu deficiency can contribute to oxidative damage because of the reduction in the enzymatic activity of various Cu-dependent antioxidant enzymes such as Cu/Zn superoxide dismutase (SOD), ceruloplasmin and cytochrome c oxidase, which play a key role in the removal of reactive oxygen species (ROS) [35]. Hence, Cu reduces the adverse complications of oxidative stress which play a major role in diabetes progression [36]. Also, it contributes to the maintenance of vascular homeostasis and the regulation of blood pressure by inhibiting angiotensin-converting enzyme (ACE) activity, which is involved in the development of HTN [37]. Previous animal studies have also reported that the restriction of Cu in the diet causes cardiac disturbances and hypertension and dyslipidemias such as high cholesterol and triglycerides (TG) in addition to an alteration in the composition of low-density lipoprotein (LDL) and very-low density lipoprotein (VLDL) [38,39].

Many studies discussed the association between Zn, Cu and CVD risk factors such as lipid profile, diabetic parameters, and HTN; however, conclusions are mixed. Regarding lipid profile, some studies found a negative association between Zn and total cholesterol (TC) and LDL [40]. While others reported no association between Zn and free fatty acids [41], lipid status [42], and CVD risk [43,44]. On the other hand, plasma Cu was positively associated with high plasma TC, high density lipoproteins (HDL), and high risk of elevated TC and LDL dyslipidemia [45]. Additionally, an inverse relationship was observed between Cu/Zn and TC/HDL, LDL/HDL, and non-HDL/HDL ratios, respectively [41]. Another study did not find any differences in HbA1c levels and Homeostasis Model Assessment Insulin Resistance (HOMA-IR) scores across Zn groups in patients with well-controlled diabetes [46]. However, a study by Farooq et al. [47] showed that a low Zn level was associated with poor glycemic control in T2DM (negative correlation with HbA1c and fasting blood glucose) [47]. Levels of HbA1c were found to be positively associated with plasma concentration of Cu and negatively with Zn/Cu ratio in T2DM [48]. The association between diabetes and Cu levels may be related to the polymorphism of Cu/Zn SOD (also known as SOD1) [49]. Furthermore, Cu and Zn intakes and Cu/Zn ratio had no association with HTN [50,51]. However, a study showed that Cu levels higher than 130 μg/dL may elevate the susceptibility of HTN by 1.99-fold [52]. 

Several studies have suggested that the ratio of Zn to Cu, which reflects the reciprocal interaction between these two trace elements, would be a more powerful marker for the prediction and prognosis of multiple pathological states compared to individual levels of Zn and Cu [23]. For instance, high serum Cu/Zn ratio was found to be higher in patients with diabetes [53] and to be associated with increased risk of mortality from CVD [54,55], certain forms of cancer [56,57], in addition to all-cause mortality in an elderly population [58].

In the past years Qatar has gone through a major nutritional transition characterized by high levels of sedentary behavior and unhealthy eating patterns. This situation has led to higher rates of obesity and non-communicable disease (NCD) incidence. It is, therefore, important to understand factors associated with this increased CMR. In this context, no studies have investigated the impact of low trace mineral levels (Zn and Cu) and CMR among an adult population in the country. Furthermore, because of the mixed results observed in the literature, the current study was designed to evaluate the association between Zn, Cu, Zn/Cu ratio and CMR factors in a large sample size of participants from Qatar Biobank with well-characterized cardiometabolic parameters. It has been hypothesized that serum Zn and Cu levels are associated with CVD biomarkers. 

## 2. Materials and Methods

### 2.1. Study Population

This project is a cross-sectional study including Qatari and long-term residents, who have lived in Qatar for ≥15 years. Data were provided by Qatar Biobank (QBB), a platform that collects samples, health, and lifestyle information from a significant number of participants from the Qatari population. Adult men and women aged ≥ 18 years with complete data on mineral status (Zn and Cu) and cardiometabolic indicators (waist circumference (WC), body mass index (BMI), lipid profile data, serum glucose, insulin, Hb1Ac, systolic blood pressure (SBP), diastolic blood pressure (DBP), and pulse rate (PR) were included in the study. Exclusion criteria included subjects diagnosed with NCD (Diabetes, CVD, HTN), taking mineral supplements or medications, having pacemakers, pregnant or lactating women. The QBB management team identified a total of 575 adults from the master database for whom Cu and Zn levels had been measured. One hundred thirty-eight samples were excluded for not meeting the inclusion criteria. The analyzed sample consisted of 437 adults who have signed a written informed consent for using their data. The study received ethical approval from the Institutional Review Board (IRB) of QBB (Ex-2020-RES-ACC-0215-0125). 

### 2.2. Anthropometric and Biochemical Measurements

All data were collected at the QBB clinic by professional technicians and trained nurses. Weight and height were measured with light clothes and no shoes on using a calibrated scale and a wall mounted stadiometer (Seca, Hamburg, Germany), respectively. Waist circumference was measured above the iliac crest at the level of the umbilicus using a non-stretchable tape (Seca, Hamburg, Germany). Blood pressure measurements (systolic and diastolic) were taken as the average of three repeated measurements using mercury sphygmomanometer. The pulse rate was determined using a digital method. Blood samples were collected from each participant after an overnight fasting. Fasting glucose, HbA1c, insulin, total cholesterol (TC), high-density lipoprotein (HDL) cholesterol and triglycerides (TG) were determined using standard automated laboratory methods at the clinical chemistry laboratories of Hamad Medical Corporation (HMC) using Hitachi-917 (Gmbh Diagnostic, Mannheim, Germany). Friedewald formula was used to calculate LDL [59]. TG to glucose ratio (TyG) was calculated using the following formula: Ln [TG (mg/dL) × glucose (mg/dL)/2] [60,61]. Insulin levels were assayed by enzyme linked immunosorbent assay (ELISA) kit (Mercodia Insulin ELISA, Uppsala, Sweden). For insulin resistance, HOMA-IR score was calculated using the following formula: fasting glucose (mg/dL) × fasting insulin (μU/mL)/405 [62]. 

### 2.3. Zinc and Copper Measurements in Plasma

Total Cu and Zn were measured in plasma by Inductively coupled plasma mass spectrometry (ICP-MS) with external calibration at the clinical chemistry laboratories of Hamad Medical Corporation (HMC). Briefly, plasma samples (50 µL) were diluted 1:100 in a diluent containing; Triton-X-100 (0.05%), ammonium hydroxide (NH_4_OH) (1%), ethylenediaminetetraacetic Acid (EDTA; 0.05%), and Butanol (2%) (Sigma-Aldrich, Taufkirchen, Germany). Butanol was added to compensate for differences in carbon content between the samples and standards that may cause variances in ionization efficiency leading to erroneous results [63,64]. Rhodium (Rh) was used as an internal standard for accuracy to correct for any instrumental drift. Calibration standards for Cu (cat No. ICP-129; Agilent) and Zn (cat No. ICP-130; Agilent) at a concentration of 10,000 µg/mL were used to establish an analytic measurement range of 0–5000 ng/mL. Samples and standards were directly submitted to a ThermoScientific iCAP Q ICP-MS (ThermoScientific, Waltham, MA, USA) for analysis. For quality control purposes and to ensure accuracy of methods used, Seronorm trace elements serum level 1 (cat No. 201405) and level 2 (cat No. 203105) (Sero AS, Norway) were used as certified standard reference material. These are human serum accuracy control materials for the analyses of trace elements. Lyophilized reference material was reconstituted according to the supplier recommendations and analyzed with each calibration and batch of sample analysis. 

### 2.4. Statistical Analysis

Analysis of data was done using IBM SPSS Statistics, version 26 (IBM Corporation, Armonk, New York, NY, USA). Descriptive statistics were used for continuous (means and standard deviation) and categorical (frequencies, percentages) data to summarize the sample population’s demographics. Multiple linear regression was performed to determine the relationship between Zn, Cu and Zn/Cu tertiles and CMR. Odds ratio and 95% confidence intervals (CI) were determined. The regression model was adjusted for gender, age, and BMI. Partial correlation was used to elucidate the association between mineral indicators (Z, Cu, and Zn/Cu ratio) and cardiometabolic factors. CMR factors were identified based on the National Cholesterol Education Program (NCEP) Adult Treatment Panel III (ATP III) criteria: insulin resistance/diabetes (HbA1c ≥ 6.1%; HOMA-IR ≥ 2.28 [65]; TyG ≥ 8.65 [66]), lipid profile (TG > 1.7 mmol/L; LDL-C > 3.5 mmol/L; TC > 5.2 mmol/L [65]; HDL-C < 1.04 mmol/L (male) and HDL-C < 1.29 mmol/L (female)), PR > 90 bpm, and BP (SBP > 130 mmHg; DBP > 85 mmHg [61]). Metabolic syndrome (MetS) was characterized by having 3 out of the following 5 criteria; high fasting blood glucose (≥110 mg/dL) or on diabetic treatment, high BP (SBP ≥ 130 mmHg and/or DBP ≥ 85 mmHg) or on HTN treatment, high TG (≥150 mg/dL), low HDL (˂40 mg/dL in males and ˂50 mg/dL in females), and central obesity/adiposity (characterized by WC ˃ 102 cm in males and ˃88 cm in females) [67,68]. Zn deficiency was defined as plasma Zn < 10.1 µmol/L; Cu deficiency was defined as plasma Cu < 11 µmol/L [69].

## 3. Results

Table 1 shows the participants’ baseline characteristics. The mean age of the 437 participants was 41 ± 1.6 years, 62.8% were females, 89.7% were Qatari, and less than half of them had higher education (43.6%). The mean serum Zn and Cu levels were 12.6 ±2.0 and 18.2 ±4.7 µmol/L, respectively. The mean BMI was 29.98 ± 6.08 Kg/m^2^ and nearly half of the study population were obese (45.9%) while approximately one third were overweight (33.7%). The mean PR was 74 ± 10.1 bpm, the mean DBP was 66.8 ± 10.8 mm Hg and the mean SBP was 114.3 ± 16.1 mm Hg. Both average pulse rate and blood pressure (SBP/DBP) were within the normal levels (60–100 bpm and 90/60–120/80 mmHg, respectively). Moreover, the mean HbA1C was 5.7 ± 1.2%, and 13.1% of participants were considered prediabetic. The percentages of participants with high HOMA-IR and high TyG were remarkably high (48.5% and 59.6%, respectively). Furthermore, 28.7% had high cholesterol, 31.7% had low HDL, 26.5% had high LDL, and 21.1% had high TG. A proportion of 9.4% and 12.4% of the study population were Zn and Cu deficient, respectively. 

Cardiometabolic indicators according to serum Zn, serum Cu, and Zn to Cu ratio are presented in Table 2. Results indicated that only TG and TyG were significantly different between Zn tertiles, *p* ˂ 0.05 for both. An increase in serum TG and TyG with an increase in serum Zn was noted. It has also been observed a significant increase in BMI (*p* ˂ 0.001), age (*p* ˂ 0.05), and PR (*p* ˂ 0.05) with an increase in serum Cu. Participants in the highest Cu tertile had the highest TC (*p* = 0.01), HDL (*p* ˂ 0.001), and HbA1c (*p* ˂ 0.01). Furthermore, the age and BMI significantly decreased with an increase in Zn/Cu ratio, *p* ˂ 0.001 for both. A similar pattern was also noted for TC (*p* ˂ 0.05) and HDL (*p* ˂ 0.001). For TG, the maximum concentration was observed among participants in Table 2 (*p* ˂ 0.05).

Table 3 represents the partial correlation between CMR indicators and trace minerals. WC (0.10; *p* ˂ 0.05), TG (0.10; *p* ˂ 0.05), and TyG (0.12; *p* ˂ 0.05) were positively correlated with Zn. On the other hand, Cu level was positively correlated with BMI (0.23; *p* ˂ 0.001), PR (0.18; *p* ˂ 0.001), TC (0.13; *p* = 0.01), HDL (0.27; *p* ˂ 0.001), and HbA1c (0.10; *p* ˂ 0.05), and negatively with DBP (−0.13; *p* = 0.01). A positive relationship was found between Zn/Cu ratio and WC (0.13; *p* = 0.01), SBP (0.13; *p* ˂ 0.01), and DBP (0.14; *p* ˂ 0.01). In contrast to Cu levels alone, BMI (−0.19; p˂0.001), PR (−0.17; *p* ˂ 0.001), TC (−0.10; *p* ˂ 0.05), and HDL (−0.27; *p* ˂ 0.001) were negatively correlated with Zn/Cu ratio.

The association between minerals and CMR are summarized in Table 4. There were no significant associations between Zn tertiles and the studied CMR indicators (Table 4). However, high levels of Cu were significantly and negatively associated with high DBP (0.235; CI 0.056–0.986) and low HDL (0.209; CI 0.095–0.462) (Table 4). Results presented in Table 4, demonstrate that high Zn/Cu ratio (T2) was significantly associated with a decrease in the occurrence of high SBP (0.307; CI 0.098–0.968). Increased Zn/Cu ratio was significantly associated with 4.5 folds increase in low HDL (4.508; CI 2.126–9.561).

Table 5 summarizes the association between serum Zn, Cu, Zn/Cu and MetS. There was no association between Zn tertiles and the occurrence of MetS. However, results indicated that high Cu levels were significantly associated with low rate of MetS (0.121; CI 0.038–0.388). Interestingly, MetS rate increased by more than 5.5 folds with the increase in Zn/Cu ratio in T3 (5.570; 1.817–17.071).

## 4. Discussion

Zn and Cu are two essential trace minerals that are important for metabolism. Zn plays a key role in inflammation [8,9], oxidation, and lipid [10,11,12,13] and glucose metabolism [14]. Similarly, Cu also contributes to the control of inflammation and fat metabolism, in addition to its critical actions in CVD [31,32,33,34]. 

Results of the current study indicate an increase in TG and TyG levels with the increase in Zn tertiles. Similar results have been illustrated by a previous report [70]. However, a study on elderly demonstrated opposite results with a significant decrease in TG with the increase in Zn levels [40]. The discrepancy between the findings of the current study and this previous report may be explained by the higher average age of the study population by Sales et al. [40] (82.2-years-old) compared to the current study where participants were younger (41 years-old). Another study on diabetic patients reported significantly higher TG levels in low Zn intake group (among other nutrients) [71]. Samadi et al. [72] indicated that T2DM patients had 3-fold lower plasma Zn levels and significantly higher TG levels compared to controls [72]. Serum Zn levels were found to be significantly lower in T2DM patients compared with healthy controls [73]. Moreover, the current study demonstrated a significant positive correlation between serum Zn levels and TG and TyG. However, a study on elderly [40] and another on diabetic and healthy individuals [72] both reported a significant negative correlation between Zn levels and TG. Differences with the current findings may be driven by the younger age of the current study’s population and the absence of diabetes (mean HbA1C 5.7 ± 1.2 %). 

The present study also revealed a weak positive correlation between Zn levels and WC while Zaky et al. [74] reported a negative correlation in obese patients [74] although in the present study it has been observed a high prevalence of overweight and obesity (79.6%) among participants with abdominal obesity affecting almost 35% of the participants. This discrepancy may be driven by the small sample size in the study by Zaky et al. [74] (24 obese individuals and 14 healthy controls) compared to the current study (437 participants). The current study also showed a tendency for increased WC with increased Zn tertiles while a previous study reported low serum Zn levels in the high WC group [75]. This study did not find a significant association between CMR factors and Zn levels in the multivariable adjusted model. Differences are possibly a reflection of the small sample size in this previous report [75]. Previous animal studies have suggested that low serum Zn levels increases cholesterol concentration by increasing the production of phospholipids [76] because Zn has an essential role in expressing lipid metabolism-related regulatory enzymes [77]. Moreover, Zn deficiency increases acetyl co-enzyme A (CoA) carboxylase and fatty acid synthase and decreases lipoprotein lipase [78]. Furthermore, animal studies on Zn-deficient diabetic mice demonstrated a reduction in peroxisome proliferator-activated receptor (PPAR) family activity that is involved in the regulation of fatty acids and glucose metabolism [79,80]. In addition, Zn can mimic the effects of insulin and inhibit the release of free fatty acids [81,82]. 

Furthermore, the current study reported an increase in BMI, PR, TC, HDL, and HbA1c with the increase in Cu circulating levels. Similar results were indicated for age, BMI, TC, and HDL in another study [45]. Cu level was reported to be significantly higher in hypertensive patients [50,52]. Moreover, the current study found a negative correlation between Cu levels and DBP. The opposite was, however, seen in another study which compared Cu levels in non-hypertensive individuals to hypertensive patients [52]; however, in the current study individuals with established hypertension were not included. A non-significant positive association between Cu levels and DBP was also found in a study done on adolescents [83]. The results of this study indicated that an increase in Cu levels was significantly associated with decreased rate of high DBP. A previous study found that, when adjusting for sex and age, high serum Cu levels was significantly associated with HTN [50]. In addition, another study indicated that both high and low serum Cu levels were not significantly associated with HTN [51]. Yao et al. [84] did not reveal an association between high Cu intake and high BP in children and adolescents [84]. These discrepancies with the current study may be driven by the exclusion of individuals with established hypertension. The link between serum Cu levels and HTN is controversial [85], however, increased cardiac output and BP might be explained by the reduction in hemoglobin and the consequential anemia that result due to Cu deficiency [86]. Cu protects against the buildup of islet amyloid peptide which is a major amyloid deposit in the β-cells of T2DM patients [87,88].

The present study also reported a positive correlation between serum Cu levels and HbA1c. Other studies have revealed either a non-significant negative [72] or a positive [89] correlation. Consistent with the current observations, a significant positive correlation was found in diabetic patients [48] and obese people [90]. Furthermore, a positive correlation was found between Cu levels and BMI in the current study. Yang et al. demonstrated a similar result among obese people [90] while Samadi et al. [72] reported a non-significant correlation in diabetic patients [72]. 

Cu level was also positively correlated with TC and HDL in this Qatari study. Previous studies did not find a significant correlation between serum Cu levels and TC and HDL [72,90]; however, the sample size in these previous studies was smaller to the current study. In adolescents, TC showed a significant positive correlation with serum Cu [83]. The current study also reported that the increase in Cu level was significantly associated with a decrease in the prevalence of low HDL. A cohort study found no association between low and high HDL and Cu when adjusted for sex and BMI [91]. Also, no association was found between low HDL and serum Cu levels after adjustment for multiple variables in another study [45]. Cu deficiency can cause damage to cardiovascular morphology and physiology due to Cu-dependent enzyme variation, peroxidation as a result of free radical accumulation due to distress to the Cu antioxidant effect, protein glycation due to glucose accumulation and abnormal carbohydrate metabolism, and impaired protein structure and function [92]. The relationship between Cu levels and lipid profile could be because of Cu on catalase and glutathione peroxidase, two enzymes that reduce hydrogen peroxidase and modulate oxidative stress and redox-mediated responses [93]. High serum Cu level inhibits glutathione reductase and reduces glutathione production, thus, altering lipid metabolism [94,95]. Furthermore, SOD activity is decreased with Cu deficiency which consequently increases the synthesis of hydroxyl free radicals that have a central link to atherosclerosis [96]. DiSilvestro et al. [97] have demonstrated that high blood cholesterol, BP, and blood homocysteine; elevated arterial damage, glucose intolerance, oxidative damage, and thrombosis can be caused by Cu deficiency which results in decreased synthesis of dehydroepiandrosterone [97]. An animal study has demonstrated the increase in the activity of 3-hydroxy-3-methyl-glutaryl (HMG) CoA reductase with Cu deficiency, which in that study, had resulted in hypercholesterolemia [98]. 

Previous epidemiological studies have reported an association between serum Cu and Zn level imbalances and the risk of diabetes and CVD [99,100,101,102,103,104,105]. However, there is an insufficient amount of information and evidence linking Zn/Cu ratio with CMR factors. Therefore, in this study, it has been focused on this important variable which reflects the reciprocal interaction between these two trace elements. The average Zn/Cu reported in the present study was similar to that reported by Hamasaki et al. (0.69 ± 0.23) [106] and lower than that of Samadi et al. (4.59 ± 1.11) [72]. Results of the current study demonstrated a significant increase in Zn/Cu ratio with lower age, BMI, TC, and HDL. Although significant, TG levels did not change between the first and third tertiles (T3) but increased in the second (T2). A previous study indicated lower Zn/Cu ratio in diabetic patients when compared to healthy controls [72]. In the same study, age, BMI, TC, and TG were significantly higher in diabetic individuals [72]. Recently, Cabral et al. reported that higher serum Zn levels were associated with an increased risk of incident T2DM [107]. In the present Qatari study, TC and HDL were negatively correlated with Zn/Cu ratio. These results are in line with the results of a previous study [72]. However, a Japanese study did not report this correlation [106]. Furthermore, we found that the second Zn/Cu ratio tertile was associated with a decrease in the prevalence of high SBP and the third tertile was associated with an increased risk of having low HDL. Previously, other studies have assessed Cu/Zn ratio which was found to be significantly higher in hypertensive people compared to controls [50]. In addition, no significant association was reported between the highest serum Cu/Zn ratio and HTN [50] or the highest intake of Cu/Zn ratio and high BP [84]. Recently, it has also been observed that higher serum levels of Cu and Cu/Zn ratio were associated with a heightened risk of CVD [107]. 

MetS is a multifactorial disease characterized by glucose intolerance, HTN, android obesity, atherogenic dyslipidemia, and consequently, pro-inflammation [108]. In line with a previous report [44], the current results also indicated no association between serum Zn levels and MetS. However, previous studies reported lower Zn levels in patients with MetS [109] (males only [110]) while other studies revealed no association [111]. High serum Cu levels was found to significantly associate with lower risk of MetS in the current study. Previous studies have revealed no association between serum Cu and risk of MetS [109,110,111]. In line with the current study, Wen et al. [112], found a strong association between urinary Cu and MetS (OR: 17; 95% CI: 2.254–4.833) [112]. Another study conducted in China among adults reported an association between elevated urinary Cu and Zn and an increased rate of MetS. The same study reported an increase in CRP levels with an increase in urinary Cu and Zn, and plasma CRP was positively associated with MetS prevalence. The study concluded that systemic inflammation may be one of the possible mechanisms behind the association between Zn and Cu and MetS [113]. The relationship between Zn and Cu levels and the inflammatory markers in the pathophysiologic mechanisms associated with MetS was reported by other studies [109,114,115]. It has been observed here that high Zn/Cu ratio significantly increased the risk for MetS. Increased Zn/Cu ratio was associated with decreased risk of poor glycemic control in diabetic patients in the Japanese study [106]. 

This study should, however, be considered in light of some limitations. First and most importantly, the study design, which is cross-sectional, cannot establish a causation between the minerals and cardiometabolic markers. Second, the sample size might be too small to represent the population being studied. On the other hand, a major strength of the current study is the inclusion of multiple CMR indicators which allowed a comprehensive analysis of the association with Zn and Cu levels. 

## 5. Conclusions

In the present study, no association was found between serum Zn levels and CMR factors or MetS. Furthermore, high serum Cu ˃ 15.8 µmol/L, was associated with a protective effect on DBP, HDL and MetS. Serum Zn/Cu ratio was found to be associated with lower risk of high SBP only in the medium serum levels (0.63–0.79 µmol/L); however, serum Zn/Cu ratio > 0.79 µmol/L was associated with the risk of having low HDL. Also, Zn/Cu ratio > 0.63 µmol/L was associated with an increased risk of MetS. This study is the first of its kind in Qatar and the region and it underscores the association between trace element status, particularly Zn and Cu, and the risk of CVD and MetS. Future studies are recommended to focus on minerals status among obese, abdominally obese, and prediabetic subjects because of the probable link between low serum Zn and Cu levels and insulin resistance and CVD.

## Figures and Tables

**Table 1 nutrients-13-02729-t001:** Baseline characteristics of the study population ^1^.

Demographics	
Study population size	437
Age (years)	41.0 (12.6)
Sex (male/female) (%)	37.2%/62.8%
Nationality (Qatari/non-Qatari) (%)	89.7%/10.1%
Education level (primary/secondary/university) (%)	13.2%/43.2%/43.6%
**Cardiometabolic Factors ^2^**	
BMI (Kg/m^2^)	29.98 (6.05)
OW and OB N (%)	347 (79.6)
AO N (%)	152 (34.9)
WC (cm)	88.4 (14.5)
PR (bpm)	70.4 (10.1)
DBP (mmHg)	66.8 (10.8)
SBP (mmHg)	114.3 (16.1)
Glu (mmol/L)	5.8 (1.9)
TC (mmol/L)	4.97 (1.02)
High cholesterol N (%)	125 (28.7)
HDL (mmol/L)	1.4 (0.4)
Low HDL N (%)	138 (31.7)
LDL (mmol/L)	2.98 (0.93)
High LDL N (%)	115 (26.5)
TG (mmol/L)	1.3 (0.9)
High TG N (%)	92 (21.1)
Insulin (mIU/L)	11.97 (16.40)
HbA1C (%)	5.7 (1.2)
Prediabetes N (%)	57 (13.1)
Homocysteine (μmol/L)	9.4 (3.0)
HOMA-IR	3.5 (7.2)
High HOMA-IR N (%)	206 (48.5)
TyG	4.6 (0.32)
High TyG N (%)	260 (59.6)
Copper level (μmol/L)	18.2 (4.7)
Zinc level (µmol/L)	12.6 (2.0)
Zn to Cu ratio	0.7 (0.2)
Prevalence of Cu deficiency N (%)	54 (12.4)
Prevalence of Zn deficiency N (%)	41 (9.4)

^1^ Results are expressed as mean (SD) and N (%) for categorical variables. ^2^ BMI = body mass index; OW = overweight; OB = obesity; AO = abdominal obesity; WC = waist circumference; PR = pulse rate; DBP = diastolic blood pressure; SBP = systolic blood pressure; Glu = glucose; TC = total cholesterol; HDL = high density lipoproteins; LDL = low density lipoproteins; TG = triglycerides; HbA1c = hemoglobin A1C; HOMA-IR = homeostatic model assessment of insulin resistance; TyG = triglyceride-glucose index. Zn deficiency defined as plasma Zn < 10.1 µmol/L; Cu deficiency defined as plasma Cu < 11 µmol/L.

**Table 2 nutrients-13-02729-t002:** Comparison of cardiometabolic markers according to serum Zn, Cu, and Zn to Cu ratio ^1^.

	Zinc Tertiles				Copper Tertiles				Zn to Cu Ratio Tertiles			
	T1: <11.6 µmol/L (n = 144)	T2: 11.6–13.3 µmol/L (n = 146)	T3: >13.3 µmol/L (n = 146)	*p*-Value ^3^	T1: <15.8 µmol/L (n = 146)	T2: 15.8–19.1 µmol/L (n = 146)	T3: >19.1 µmol/L (n = 144)	*p*-Value ^3^	T1: <0.63 (n = 144)	T2: 0.63–0.79 (n = 144)	T3: >0.79 (n = 148)	*p*-Value ^3^
Cardiometabolic Markers ^2^												
Age (years)	40.9 (12.4)	41.6 (12.1)	40.6 (13.2)	0.760	38.9 (13.0)	42.0 (13.0)	42.2 (11.4)	**0.043**	41.8 (11.1)	43.6 (12.9)	37.9 (12.9)	**0.000**
BMI (Kg/m^2^)	30.3 (6.1)	29.6 (6.3)	30.0 (5.8)	0.594	28.0 (5.5)	29.8 (5.7)	32.1 (6.3)	**0.000**	31.4 (6.0)	30.4 (6.0)	28.2 (5.8)	**0.000**
WC (cm)	87.0 (14.1)	88.5 (15.7)	89.8 (13.5)	0.252	88.2 (15.1)	88.0 (13.9)	89.2 (14.5)	0.754	86.8 (13.1)	89.8 (15.3)	88.7 (14.9)	0.201
PR (bpm)	71.3 (10.6)	70.3 (9.6)	69.7 (10.1)	0.387	68.7 (10.2)	71.1 (9.8)	71.5 (10.2)	**0.040**	71.7 (10.7)	70.8 (8.9)	68.9 (10.5)	0.052
DBP (mmHg)	65.7 (10.5)	67.8 (11.9)	66.9 (9.9)	0.274	68.1 (12.0)	66.1 (9.3)	66.2 (10.9)	0.198	65.9 (10.5)	66.0 (10.6)	68.5 (11.2)	0.076
SBP (mmHg)	112.8 (15.6)	114.9 (17.1)	115.0 (15.6)	0.431	114.6 (15.0)	113.3 (16.3)	114.9 (17.0)	0.666	113.9 (16.8)	114.1 (16.8)	114.8 (14.7)	0.887
Glu (mmol/L)	5.6 (1.9)	5.9 (2.1)	5.8 (1.7)	0.607	5.5 (1.4)	5.8 (1.8)	6.0 (2.3)	0.086	5.8 (2.1)	5.9 (2.0)	5.6 (1.6)	0.334
TC (mmol/L)	4.8 (0.9)	5.1 (1.1)	5.0 (1.0)	0.101	4.8 (1.0)	5.1 (1.1)	5.1 (1.0)	**0.010**	5.0 (0.9)	5.1 (1.1)	4.8 (1.0)	**0.018**
HDL (mmol/L)	1.5 (0.4)	1.4 (0.4)	1.4 (0.4)	0.136	1.3 (0.3)	1.4 (0.4)	1.5 (0.4)	**0.000**	1.5 (0.4)	1.4 (0.4)	1.3 (0.4)	**0.000**
LDL (mmol/L)	2.9 (0.9)	3.1 (1.0)	3.0 (0.9)	0.127	2.9 (1.0)	3.0 (0.9)	3.0 (0.9)	0.439	3.0 (0.8)	3.1 (1.0)	2.9 (1.0)	0.380
TG (mmol/L)	1.1 (0.6)	1.4 (1.4)	1.3 (0.6)	**0.039**	1.2 (0.6)	1.3 (1.2)	1.3 (0.8)	0.823	1.2 (0.6)	1.4 (1.4)	1.2 (0.6)	**0.031**
Insulin (mIU/L)	13.9 (21.5)	10.4 (5.6)	11.6 (17.8)	0.185	12.5 (19.5)	10.8 (10.1)	12.7 (18.1)	0.560	12.7 (19.6)	13.2 (19.7)	10.1 (6.3)	0.233
HbA1c (%)	5.6 (1.2)	5.7 (1.2)	5.8 (1.2)	0.259	5.4 (0.8)	5.8 (1.3)	5.9 (1.4)	**0.001**	5.8 (1.3)	5.8 (1.2)	5.5 (1.1)	0.098
HOMA-IR	4.1 (8.1)	2.9 (2.6)	3.5 (9.2)	0.360	3.5 (7.2)	3.0 (3.9)	4.0 (9.4)	0.557	3.9 (9.7)	3.9 (7.3)	2.7 (2.9)	0.275
TyG	4.5 (0.3)	4.6 (0.4)	4.6 (0.3)	**0.033**	4.6 (0.3)	4.6 (0.3)	4.6 (0.3)	0.460	4.6 (0.3)	4.6 (0.4)	4.6 (0.3)	0.121

^1^ Results are expressed as Mean (SD). ^2^ WC = waist circumference; PR = pulse rate; DBP = diastolic blood pressure; SBP = systolic blood pressure; Glu = glucose; TC = total cholesterol; HDL = high density lipoproteins; LDL = low density lipoproteins; TG = triglycerides; HbA1c = hemoglobin A1C; HOMA-IR = homeostatic model assessment of insulin resistance; TyG = triglyceride-glucose index. ^3^ Bold indicates statistically significant results. There were no differences observed in Zn, Cu levels and Zn/Cu ratio between participants with and without MetS (results not shown).

**Table 3 nutrients-13-02729-t003:** Partial Correlation between cardiometabolic indicators and trace minerals (Zn, Cu, and Zn/Cu) ^1^.

	Zn		Cu		Zn/Cu	
Biomarkers	r	*p*-Value ^2^	r	*p*-Value ^2^	r	*p*-Value ^2^
BMI (Kg/m^2^)	−0.02	0.732	0.23	**<** **0.001**	−0.19	**<** **0.001**
WC (cm)	0.10	**0.037**	−0.07	0.186	0.13	**0.010**
SBP (mmHg)	0.09	0.060	−0.06	0.194	0.13	**0.007**
DBP (mmHg)	0.05	0.309	−0.13	**0.010**	0.14	**0.003**
PR (bpm)	−0.05	0.268	0.18	**<** **0.001**	−0.17	**<** **0.001**
Glu (mmol/L)	0.03	0.544	0.01	0.876	0.00	0.997
TC (mmol/L)	0.03	0.547	0.13	**0.010**	−0.10	**0.038**
HDL (mmol/L)	−0.09	0.065	0.27	**<** **0.001**	−0.27	**<** **0.001**
LDL (mmol/L)	0.04	0.396	0.01	0.779	−0.01	0.799
TG (mmol/L)	0.10	**0.034**	0.01	0.844	0.08	0.123
Insulin (mIU/L)	−0.06	0.213	0.01	0.832	−0.05	0.320
HbA1c (%)	0.08	0.095	0.10	**0.038**	−0.05	0.357
HOMA-IR	−0.05	0.322	0.02	0.766	−0.05	0.342
TyG	0.12	**0.012**	0.02	0.671	0.07	0.139

^1^ r = correlation coefficient; WC = waist circumference; PR = pulse rate; DBP = diastolic blood pressure; SBP = systolic blood pressure; Glu = glucose; TC = total cholesterol; HDL = high density lipoproteins; LDL = low density lipoproteins; TG = triglycerides; HbA1c = hemoglobin A1C; HOMA-IR = homeostatic model assessment of insulin resistance; TyG = triglyceride-glucose index. ^2^ Bold indicates statistically significant results.

**Table 4 nutrients-13-02729-t004:** Association between serum Zn Cu Zn/Cu and cardiometabolic markers ^2,3^.

	Zn Tertiles						
	T1 ^1^	T2			T3		
	AOR	AOR	95% CI	*p*-Value ^4^	AOR	95% CI	*p*-Value ^4^
High SBP (mmHg)	Reference	0.999	0.327–3.056	0.999	1.351	0.446–4.099	0.595
High DBP (mmHg)	Reference	1.340	0.356–5.044	0.666	0.634	0.134–3.009	0.566
High PR (bpm)	Reference	0.795	0.085–7.466	0.841	0.000	0.000	0.996
High TC (mmol/L)	Reference	1.335	0.647–2.756	0.435	1.545	0.754–3.166	0.235
Low HDL (mmol/L)	Reference	1.405	0.706–2.795	0.333	1.090	0.544–2.184	0.809
High LDL (mmol/L)	Reference	1.650	0.789–3.450	0.183	1.384	0.654–2.928	0.395
High TG (mmol/L)	Reference	2.130	0.896–5.063	0.087	1.681	0.689–4.102	0.254
High HbA1c (%)	Reference	0.974	0.370–2.565	0.958	1.162	0.443–3.053	0.760
High HOMA-IR	Reference	1.690	0.854–3.346	0.132	1.128	0.569–2.234	0.730
High TyG	Reference	1.558	0.789–3.074	0.201	1.635	0.823–3.246	0.160
	**Cu Tertiles**						
	**T1 ^1^**	**T2**			**T3**		
	**AOR**	**AOR**	**95% CI**	***p*** **-Value ^4^**	**AOR**	**95% CI**	***p*** **-Value ^4^**
High SBP (mmHg)	Reference	0.416	0.130–1.328	0.139	0.659	0.224–1.943	0.450
High DBP (mmHg)	Reference	0.074	0.009–0.635	**0.018**	0.235	0.056–0.986	**0.048**
High PR (bpm)	Reference	0.364	0.016–8.481	0.529	1.628	0.048–54.84	0.786
High TC (mmol/L)	Reference	1.348	0.647–2.808	0.425	1.767	0.842–3.706	0.132
Low HDL (mmol/L)	Reference	0.474	0.240–0.935	**0.031**	0.209	0.095–0.462	**<** **0.001**
High LDL (mmol/L)	Reference	1.123	0.541–2.330	0.755	1.124	0.529–2.389	0.761
High TG (mmol/L)	Reference	0.651	0.291–1.455	0.295	0.426	0.178–1.016	0.054
High HbA1c (%)	Reference	0.655	0.238–1.803	0.413	0.991	0.379–2.590	0.985
High HOMA-IR	Reference	0.733	0.368–1.460	0.378	0.724	0.355–1.477	0.375
High TyG	Reference	0.768	0.388–1.523	0.451	0.722	0.352–1.478	0.373
	**Zn/Cu Tertiles**						
	**T1 ^1^**	**T2**			**T3**		
	**AOR**	**AOR**	**95% CI**	***p*** **-Value ^4^**	**AOR**	**95% CI**	***p*** **-Value ^4^**
High SBP (mmHg)	Reference	0.307	0.098–0.968	**0.044**	1.039	0.361–2.991	0.943
High DBP (mmHg)	Reference	0.635	0.133–3.016	0.567	1.575	0.410–6.046	0.508
High PR (bpm)	Reference	1.224	0.088–17.066	0.881	0.000	0.000	0.996
High TC (mmol/L)	Reference	0.992	0.497–1.981	0.981	0.785	0.384–1.602	0.505
Low HDL (mmol/L)	Reference	1.736	0.798–3.773	0.164	4.508	2.126–9.561	**<** **0.001**
High LDL (mmol/L)	Reference	1.135	0.548–2.353	0.733	1.199	0.579–2.483	0.626
High TG (mmol/L)	Reference	1.834	0.782–4.305	0.163	2.014	0.843–4.810	0.115
High HbA1c (%)	Reference	0.727	0.278–1.900	0.515	1.187	0.458–3.074	0.724
High HOMA-IR	Reference	1.579	0.790–3.156	0.196	1.519	0.761–3.032	0.236
High TyG	Reference	1.248	0.623–2.498	0.532	1.690	0.842–3.392	0.140

^1^ T1: reference category. ^2^ Adjusted for: gender, age, BMI, and physical activity. ^3^ AOR = adjusted odds ratio; WC = waist circumference; PR = pulse rate; DBP = diastolic blood pressure; SBP = systolic blood pressure; Glu = glucose; TC = total cholesterol; HDL = high density lipoproteins; LDL = low density lipoproteins; TG = triglycerides; HbA1c = hemoglobin A1C; HOMA-IR = homeostatic model assessment of insulin resistance; TyG = triglyceride-glucose index. ^4^ Bold indicates statistically significant results.

**Table 5 nutrients-13-02729-t005:** Association between serum Zn, Cu, and Zn/Cu and the risk of having MetS ^1,3^.

	Adjusted OR	95% CI	*p*-Value ^4^
Zn tertiles			
T1 ^2^	1	-	-
T2	1.269	0.500–3.219	0.616
T3	1.299	0.510–3.310	0.583
Cu tertiles			
T1 ^2^	1	-	-
T2	0.508	0.213–1.211	0.127
T3	0.121	0.038–0.388	**<** **0.001**
Zn to Cu ratio tertiles			
T1 ^2^	1	-	-
T2	3.870	1.272–11.773	**0.017**
T3	5.570	1.817–17.071	**0.003**

^1^ MetS is characterized by having 3 out of the 5 indicators, high blood glucose, high blood pressure, high TG, low HDL, and central obesity/adiposity [67]. ^2^ T1: reference category. ^3^ Adjusted for: gender, age, BMI, and physical activity. ^4^ Bold indicates statistically significant results.

## Data Availability

The data set generated for the study is available on request to Qatar Biobank study management team.

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
