# Peer review of "The Association between Zinc and Copper Circulating Levels and Cardiometabolic Risk Factors in Adults: A Study of Qatar Biobank Data"

_nutrients, 2021, doi:10.3390/nu13082729_

Round 1

Reviewer 1 Report

In their manuscript „The Association Between Zinc and Copper Circulating  Levels and Cardiometabolic Risk Factors in Adults: A  Study of Qatar Biobank Data” the authors present an association study using pre-existing data from a biobank.

First and foremost, I have a problem understanding the selection process of patients to be included (“simple random selection”). If I understood correctly, no additional measurements were required for the present manuscript, only pre-measured data from a databased were analyzed (inclusion criteria were a complete set of all relevant data). In this case, it is an acceptable workload to include all patients in the database meeting the inclusion criteria, instead of just a representative sub-sample. This increases the statistical power, which is certainly recommended for a study which is just based on less than 500 samples (as mentioned in line 346). So, how many eligible patients are in the Qatar Biobank in total? This should be the number of participants.

Specific points

Please do always consider that an association study, by design, is unable to deduce any causal relationships. In the present case, any correlation between zinc and copper with a physiological parameter does not necessarily mean that there is an effect of the metal on said parameter. It is also very well possible that a certain state of disease, or a physiological parameter, affects metal ion homeostasis, leading to an association. Therefore, all conclusions about causal relationships cannot be answered by this study design and need to be avoided in the manuscript. This starts in the abstract (and should be extended consequently through the rest of the manuscript), where the authors end with two unjustified conclusions:

“Our finding suggest that high plasma Cu had a protective effect on DBP, HDL and MetS.” This is not necessarily the case. It might also be that there is a common underlying cause affecting Cu as well as DBP, HDL and MetS, or that DBP, HDL and MetS affect Cu plasma levels.

“While high plasma Zn/Cu ratio increased the risk of having low HDL and MetS.” As above, there may be a common cause for all, or HDL and Met S might affect the Zn/Cu ration (or Cu and thereby the Zn/Cu ratio). In addition, please not that starting this sentence with “While” is an awkward phrasing.

Please correct these and all other incidences of unjustified claims of causal relationships throughout the manuscript, especially the conclusions. As correctly stated in line 344-346 this type of analysis is limited to detecting associations. Importantly, it is not sufficient to mention this as a limitation, but then proceed to make causal conclusions anyway.

Just to be clear: Have the patients been pre-selected by any other criteria, such as adipositas? An average BMI of 29.98 seems rather high for a general population.

Line 206: I would recommend to change “trace minerals” to “Zn and Cu”. As there are many other trace elements with a potential relevance in CVR this study is limited to only two of them.

There is some relevant recent literature that is missing from the discussion, e.g., a study with 2,500 participants were, among others, the associations of Zn and Cu with CVD were investigated (Cabral et al. 2021 Eur J Nutr 10.1007/s00394-021-02494-3) and references therein.

Line 310: Here, a causal effect of Zn and Cu on diabetes and CVD is postulated for other studies, which were epidemiological by design. As for the present study, please do not conclude any causal relationships where this is not justified.

Author Response

Reviewer 1

We thank the reviewer for his/her critical review of this manuscript. The pertinent comments and suggestions of the reviewer greatly helped us enhancing the quality of the manuscript. All changes are highlighted in the revised version of the manuscript.

In their manuscript „The Association Between Zinc and Copper Circulating Levels and Cardiometabolic Risk Factors in Adults: A Study of Qatar Biobank Data” the authors present an association study using pre-existing data from a biobank.

First and foremost, I have a problem understanding the selection process of patients to be included (“simple random selection”). If I understood correctly, no additional measurements were required for the present manuscript, only pre-measured data from a databased were analyzed (inclusion criteria were a complete set of all relevant data). In this case, it is an acceptable workload to include all patients in the database meeting the inclusion criteria, instead of just a representative sub-sample. This increases the statistical power, which is certainly recommended for a study which is just based on less than 500 samples (as mentioned in line 346). So, how many eligible patients are in the Qatar Biobank in total? This should be the number of participants.

Response: In the current study, we have requested from Qatar Biobank (QBB) all participants who had Cu and Zn levels measured. By interrogating QBB master database, 575 subjects were identified for whom Cu and Zn data were available. Among them, 437 participants met our inclusion/exclusion criteria and those were included into the analysis for this study. This point has now been better clarified in the methods section. Changes are highlighted in the revised version of the manuscript.

Specific points

Please do always consider that an association study, by design, is unable to deduce any causal relationships. In the present case, any correlation between zinc and copper with a physiological parameter does not necessarily mean that there is an effect of the metal on said parameter. It is also very well possible that a certain state of disease, or a physiological parameter, affects metal ion homeostasis, leading to an association. Therefore, all conclusions about causal relationships cannot be answered by this study design and need to be avoided in the manuscript. This starts in the abstract (and should be extended consequently through the rest of the manuscript), where the authors end with two unjustified conclusions:

“Our finding suggest that high plasma Cu had a protective effect on DBP, HDL and MetS.” This is not necessarily the case. It might also be that there is a common underlying cause affecting Cu as well as DBP, HDL and MetS, or that DBP, HDL and MetS affect Cu plasma levels.

“While high plasma Zn/Cu ratio increased the risk of having low HDL and MetS.” As above, there may be a common cause for all, or HDL and Met S might affect the Zn/Cu ration (or Cu and thereby the Zn/Cu ratio). In addition, please not that starting this sentence with “While” is an awkward phrasing.

Response: Many thanks to the reviewer for this suggestion. These 2 sentences have now been rephased taking into account the reviewer feedback. These 2 sentences read now as follows: “These findings suggest that high plasma Cu levels are associated with a protective effect on DBP, HDL and MetS and that high plasma Zn/Cu ratio increases the risk of having low HDL and MetS”. Changes are highlighted in the revised abstract section.

Please correct these and all other incidences of unjustified claims of causal relationships throughout the manuscript, especially the conclusions. As correctly stated in line 344-346 this type of analysis is limited to detecting associations. Importantly, it is not sufficient to mention this as a limitation, but then proceed to make causal conclusions anyway.

Response: As suggested by the reviewer, all reference to a causal relationship was removed throughout the text to avoid overstatement and stay within the remit of an association study.

Just to be clear: Have the patients been pre-selected by any other criteria, such as adipositas? An average BMI of 29.98 seems rather high for a general population.

Response: We did not use BMI as a selection criterion. The prevalence of overweight and obesity is very high in Qatar (over 70% are overweight) (https://www.qatarbiobank.org.qa/research/key-findings).

Line 206: I would recommend to change “trace minerals” to “Zn and Cu”. As there are many other trace elements with a potential relevance in CVR this study is limited to only two of them.

 Response: Trace minerals were changed to Zn and Cu.

There is some relevant recent literature that is missing from the discussion, e.g., a study with 2,500 participants were, among others, the associations of Zn and Cu with CVD were investigated (Cabral et al. 2021 Eur J Nutr 10.1007/s00394-021-02494-3) and references therein.

Response: As suggested by the reviewer, this very recent and highly relevant study has now been included into the discussion of the revised version of the manuscript. Changes are highlighted in the new text,

Line 310: Here, a causal effect of Zn and Cu on diabetes and CVD is postulated for other studies, which were epidemiological by design. As for the present study, please do not conclude any causal relationships where this is not justified.

Response: As suggested by the reviewer, this sentence has now been rephrased to avoid any causal relationships.

Reviewer 2 Report

This manuscript is well written and I enjoyed reading it. However, to improve the quality of the manuscript, I would like to make the following recommendations:

  1. Abstract - good representation of the study

2. Introduction

  • Although the introduction is long, it gives a thorough explanation of how all the CMR and plasma Cu and Zn levels are interrelated. It thus provides a good background and rational for the study
  • Line 50 - replace "Zrt/Irt-like (Zip)-7" with "Zrt/Irt-like protein-7 (ZIP7)"
  • Line 50 - replace "drove" with "drive
  • Line 72 - revise as follows: "... enzymes such as Cu/Zn..."
  • Line 83 - Replace "in highlight to" with "with regard to" lipid profile...
  • Line 85 - fatty acids - which fatty acids are referred to here?
  • Line 88 - include highlighted part as follows: "..., and non-HDL/HDL ratios respectively [40]."
  • Line 91 - at an "a" before "low Zn level"
  • Line 95 - write out SOD first before the abbreviation is used - superoxide dismutase
  • Line 97 - replace "folds" with "fold"
  • Line 103 - ad "an" before "elderly population".
  • Line 104 - add highlighted part and delete "a" as follows: " ... characterized by high levels of sedentary behavior..." 
  • Line 105 - revise to read as follows: "... rates of obesity and non-communicable disease (NCD) incidence."
  • Line 106 - revise to read as follows: "... with this increased CMR."
  • Line 108 - add "an" before "adult population"
  • Line 111 - revise to read as follows: "... serum Zn and Cu levels associated ..."
  • Line 111 - you refer to serum here, but in line 147 you indicate plasma Cu and Zn levels - check and be consistent throughout the manuscript.
  •  
  • 3 Materials and Methods
  • Line 116 and 131 -  "data were" instead of "data was"
  • Line 120 - Include serum glucose
  • Simple random sampling was used - it is not clear how the sampling was done - was it not convenient sampling? 575 people had data and then 138 respondents data were not used because of not meeting the inclusion criteria. OR did you use a sample calculation formula and determined  a sample size of 575 and thus randomly chose 575 respondents? Describe. This is not clear.
  • Line 132 - revise as follows: "... height were measured with light clothes and no shoes on using a calibrated scale..."
  • Line 137 - you refer to fasting. How long? At what time were the blood samples collected? Fasting from when?
  • Line 143 - ELISA - do not use he abbreviation before it was written out Enzyme linked immunosorbent essay
  • Line 150 - write out ammonium hydroxide (Nh4OH) before abbreviation is used
  • Line 153 - write our Rhodium (Rh) before Rh is used
  • Line 164 - revise as follows: "... done using IBM SPSS Statistics, version 26 (IBM Corporation, Armonk, New York)." Delete the Statistical Package... as the name changed when SPSS was taken over by IBM
  • Line 166 - include as follows: "... categorical (frequencies, percentages)..."
  • Line 169 - physical activity is mentioned, but it was not mentioned in the methods or results
  • Line 179-180 - you need a reference for the cut-points
  •  
  • 4. Results
  • Line 182 & 184 - always report mean+-SD
  • Line 186-187 - instead of only reporting the mean PR and SBP/DBP, also report the % with HTN and abnormal PR
  • Line 195 - Table 2 instead of table 2
  • Line 198-199 - also include BMI and PR as these were also significantly higher
  • Line 216 - Tables 4 A, B and C instead of tables 4
  • Line 216 - revise to read as follows: "There were no significant associations between..."
  •  
  • 5. Discussion
  • Line 242 - replace "folds" with "fold"
  • Line 260 - write out Co-enzyme A (CoA) before using the abbreviation
  • Line 270 - revise to read: "... not included. A non-significant..."
  • Line 272 - add "an" before "increase in Cu"
  • When referring to Cu and Zn - be specific and refer to Cu levels and Zn levels throughout the text   
  • Line 273 - revise as follows: "... found that, when..."
  • Line 283 - include "a" before "non-significant"
  • Line 283 - Replace with "In line" with "Consistent"
  • Line 345 - replace "that fails to conclude" with "cannot establish"

6. Conclusions

Line 351 - replace "report" with "find"

Author Response

Reviewer #2

We thank the reviewer for her/his thorough reading of the manuscript and positive opinion on the significance of our work. We are very grateful for the pertinent comments provided which helped improving the manuscript. All changes are highlighted in the revised version of the manuscript.

This manuscript is well written and I enjoyed reading it. However, to improve the quality of the manuscript, I would like to make the following recommendations:

  1. Abstract - good representation of the study
  1. Introduction
  • Although the introduction is long, it gives a thorough explanation of how all the CMR and plasma Cu and Zn levels are interrelated. It thus provides a good background and rational for the study – Done.
  • Line 50 - replace "Zrt/Irt-like (Zip)-7" with "Zrt/Irt-like protein-7 (ZIP7)" – Done.
  • Line 50 - replace "drove" with "drive – Done.
  • Line 72 - revise as follows: "... enzymes such as Cu/Zn..." – Done.
  • Line 83 - Replace "in highlight to" with "with regard to" lipid profile... – Done.
  • Line 85 - fatty acids - which fatty acids are referred to here? – Done.
    • Response: We refer to free fatty acids – now corrected.
  • Line 88 - include highlighted part as follows: "..., and non-HDL/HDL ratios respectively [40]." – Done.
  • Line 91 - at an "a" before "low Zn level" – Done.
  • Line 95 - write out SOD first before the abbreviation is used - superoxide dismutase
    • Response: SOD was spelled out and defined earlier in line 73.
  • Line 97 - replace "folds" with "fold" – Done.
  • Line 103 - ad "an" before "elderly population". – Done.
  • Line 104 - add highlighted part and delete "a" as follows: " ... characterized by high levels of sedentary behavior..." – Done.
  • Line 105 - revise to read as follows: "... rates of obesity and non-communicable disease (NCD) incidence." – Done.
  • Line 106 - revise to read as follows: "... with this increased CMR." – Done.
  • Line 108 - add "an" before "adult population" – Done.
  • Line 111 - revise to read as follows: "... serum Zn and Cu levels associated ..." – Done.
  • Line 111 - you refer to serum here, but in line 147 you indicate plasma Cu and Zn levels - check and be consistent throughout the manuscript. – Done.
  •  
  • 3 Materials and Methods
  • Line 116 and 131 -  "data were" instead of "data was" – Done.
  • Line 120 - Include serum glucose – Done.
  • Simple random sampling was used - it is not clear how the sampling was done - was it not convenient sampling? 575 people had data and then 138 respondents data were not used because of not meeting the inclusion criteria. OR did you use a sample calculation formula and determined  a sample size of 575 and thus randomly chose 575 respondents? Describe. This is not clear.
  • Response: In the current study, we have requested from Qatar Biobank (QBB) all participants who had Cu and Zn levels measured. By interrogating QBB master database, 575 subjects were identified for whom Cu and Zn data were available. Among them, 437 participants met our inclusion/exclusion criteria and those were included into the analysis for this study. This point has now been better clarified in the methods section. Changes are highlighted in the revised version of the manuscript.
  • Line 132 - revise as follows: "... height were measured with light clothes and no shoes on using a calibrated scale..." – Done.
  • Line 137 - you refer to fasting. How long? At what time were the blood samples collected? Fasting from when?
  • Response: Participants were fasting for at least 8 hours. This information has been added to the methods section.
  • Line 143 - ELISA - do not use he abbreviation before it was written out Enzyme linked immunosorbent essay – Done.
  • Line 150 - write out ammonium hydroxide (Nh4OH) before abbreviation is used – Done.
  • Line 153 - write our Rhodium (Rh) before Rh is used – Done.
  • Line 164 - revise as follows: "... done using IBM SPSS Statistics, version 26 (IBM Corporation, Armonk, New York)." Delete the Statistical Package... as the name changed when SPSS was taken over by IBM – Done.
  • Line 166 - include as follows: "... categorical (frequencies, percentages)..." – Done.
  • Line 169 - physical activity is mentioned, but it was not mentioned in the methods or results
    • Response: We agree with reviewer. Physical activity was not included in our analysis. We have now omitted ‘physical activity” from the text.
  • Line 179-180 - you need a reference for the cut-points 
  • Response: As suggested by the reviewer, a relevant refence for reference range values used to define deficiency for Cu and Zn has now been included.

  1. Results
  • Line 182 & 184 - always report mean+-SD
    • Response: Results are now reported as mean ± SD in the text.
  • Line 186-187 - instead of only reporting the mean PR and SBP/DBP, also report the % with HTN and abnormal PR
    • Response: The % of participants with high BP and PR was very low. The study included subjects without any BP or PR problems.  
  • Line 195 - Table 2 instead of table 2 – Done.
  • Line 198-199 - also include BMI and PR as these were also significantly higher
    • Response: BMI and PR were added in the results section as suggested by the reviewer. Changes are highlighted in the new text.
  • Line 216 - Tables 4 A, B and C instead of tables 4 – Done.
  • Line 216 - revise to read as follows: "There were no significant associations between..." – Done.
  •  
  1. Discussion
  • Line 242 - replace "folds" with "fold" – Done.
  • Line 260 - write out Co-enzyme A (CoA) before using the abbreviation – Done.
  • Line 270 - revise to read: "... not included. A non-significant..." – Done.
  • Line 272 - add "an" before "increase in Cu" – Done.
  • When referring to Cu and Zn - be specific and refer to Cu levels and Zn levels throughout the text.
    • Response: As suggested by the reviewer, changes have been made throughout the text.    
  • Line 273 - revise as follows: "... found that, when..." – Done.
  • Line 283 - include "a" before "non-significant" – Done.
  • Line 283 - Replace with "In line" with "Consistent" – Done.
  • Line 345 - replace "that fails to conclude" with "cannot establish" – Done.
  1. Conclusions

Line 351 - replace "report" with "find" – Done.

Reviewer 3 Report

The manuscript would read better is the use of first person voice (e.g., "our" and "we") were eliminated from the discussion section.  Several minor grammatical errors are present but are not so numerous that they could not be addressed during production.

Author Response

Reviewer #3

The manuscript would read better is the use of first person voice (e.g., "our" and "we") were eliminated from the discussion section.  Several minor grammatical errors are present but are not so numerous that they could not be addressed during production.

We thank the reviewer for her/his positive opinion on the significance of our work. As suggested by the reviewer, we have now revised the manuscript using a passive tone throughout the text. In addition, the manuscript has gone through an additional extensive round of language and grammar revision. We hope the reviewer will now find these changes satisfactory. All changes are highlighted in the revised version of the manuscript.

Round 2

Reviewer 1 Report

I thank the authors for addressing all comments raised in my previous review. While some points were sufficiently resolved, several unsubstantiated claims have not been corrected, sometimes despite the explicit statements of the authors to the contrary. Several statements still remain to be rephrased:

Line 31-33: While the first phrase has been improved, the second one still postulates a causal connection “…high plasma Zn/Cu ratio increases the risk of having…”. It does not increase the risk, it is associated with an increased risk. This is actually a major difference!

Line 114: please add an “are” (“… serum Zn and Cu levels ARE associated…”).

Line 127: I suggest the following phrase for improved clarity: “…for whom Cu and Zn levels had been measured…”. This way, it is clear that measurements were not done in the context of the present study.

Line 321 ff: “Previous  epidemiological  studies  have  demonstrated  the  effects  of  serum  Cu  and  Zn…” . No, these studies demonstrated associations, but cannot demonstrate effects. This would require an intervention. Despite their point-by-point-reply, the authors have not made any changes here and it has not been rephrased.

Line 368: “Serum Zn/Cu ratio was found to protect against high SBP…” Again, this is not covered by the data.

Line 369:” … serum Zn/Cu ratio >0.79 µmol/L increased the risk…” Once again, this is an over-interpretation of the present study.

Line 370: “Also, Zn/Cu ratio >0.63 µmol/L increased the risk of MetS.” See above

Line 371: “the importance of trace element status, particularly Zn and Cu, achieved through an optimal diet, to reduce the incidence of CVD and MetS” This claim is totally unsubstantiated. There is an association – maybe CVD and MetS LEAD to changes in mineral homeostasis? What about that? In this case, modification of the diet would be completely useless. The authors make claims that are much more than what they can actually show by their data.

Author Response

Reviewer #1

We are very grateful to the reviewer for his/her high opinion on the importance of our work and for the additional pertinent comments and suggestions that were shared with us in this second review round. By addressing these additional comments and suggestions, we believe the quality of the manuscript has been greatly improved. All changes are highlighted in the revised version of the manuscript.

I thank the authors for addressing all comments raised in my previous review. While some points were sufficiently resolved, several unsubstantiated claims have not been corrected, sometimes despite the explicit statements of the authors to the contrary. Several statements still remain to be rephrased:

Line 31-33: While the first phrase has been improved, the second one still postulates a causal connection “…high plasma Zn/Cu ratio increases the risk of having…”. It does not increase the risk, it is associated with an increased risk. This is actually a major difference!

Response: Many thanks to the reviewer for this remark. The sentence has been reworded accordingly and it now reads as follows: “These findings suggest that high plasma Cu levels are associated with a protective effect on DBP, HDL and MetS and that high plasma Zn/Cu ratio is associated with the risk of having low HDL and MetS.” Changes are highlighted (lines 31-33).

Line 114: please add an “are” (“… serum Zn and Cu levels ARE associated…”).

Response: Corrected.

Line 127: I suggest the following phrase for improved clarity: “…for whom Cu and Zn levels had been measured…”. This way, it is clear that measurements were not done in the context of the present study.

Response: Sentence has now been reworded as suggested by the reviewer.

Line 321 ff: “Previous  epidemiological  studies  have  demonstrated  the  effects  of  serum  Cu  and  Zn…” . No, these studies demonstrated associations, but cannot demonstrate effects. This would require an intervention. Despite their point-by-point-reply, the authors have not made any changes here and it has not been rephrased.

Response: Many thanks for this pertinent remark. Based on the reviewer’s feedback, this sentence has now been rephrased as follows: “Previous epidemiological studies have reported an association between serum Cu and Zn level imbalances and the risk of diabetes and CVD”. Changes are highlighted in lines 315-316.

Line 368: “Serum Zn/Cu ratio was found to protect against high SBP…” Again, this is not covered by the data.

Line 369:” … serum Zn/Cu ratio >0.79 µmol/L increased the risk…” Once again, this is an over-interpretation of the present study.

Line 370: “Also, Zn/Cu ratio >0.63 µmol/L increased the risk of MetS.” See above

Response: These 3 sentences have now been reworded to void any causal relationship. The new section reads as follows: “In the present study, no association was found between serum Zn levels and CMR factors or MetS. Furthermore, high serum Cu, ˃15.8 µmol/L, was associated with a protective effect on DBP, HDL and MetS. Serum Zn/Cu ratio was found to be associated with lower risk of high SBP only in the medium serum levels (0.63- 0.79 µmol/L); however, serum Zn/Cu ratio >0.79 µmol/L was associated with the risk of having low HDL.” Changes ae highlighted in the text (Conclusion section; lines 362-366).

Line 371: “the importance of trace element status, particularly Zn and Cu, achieved through an optimal diet, to reduce the incidence of CVD and MetS” This claim is totally unsubstantiated. There is an association – maybe CVD and MetS LEAD to changes in mineral homeostasis? What about that? In this case, modification of the diet would be completely useless. The authors make claims that are much more than what they can actually show by their data.

Response: We thank the reviewer for this pertinent remark. As per the reviewer’s feedback, this sentence has now been toned down and the revised sentence reads as follows: “This study is the first of its kind in Qatar and the region and it underscores the association between trace element status, particularly Zn and Cu, and the risk of CVD and MetS.”. Changes are highlighted in the text (Conclusion section; lines 367-368).
